# Silicon Cantilever for Micro/Nanoforce and Stiffness Calibration

**DOI:** 10.3390/s22166253

**Published:** 2022-08-19

**Authors:** Joachim Frühauf, Eva Gärtner, Zhi Li, Lutz Doering, Jan Spichtinger, Gerd Ehret

**Affiliations:** 1SiMETRICS GmbH, Am Südhang 5, 09212 Limbach-Oberfrohna, Germany; 2Physikalisch-Technische Bundesanstalt PTB, Bundesallee 100, 38116 Braunschweig, Germany

**Keywords:** silicon cantilever, elastic bending, tactile profilometer, stylus force measurement, interferometric nano-topography measurement, surface stress

## Abstract

The paper deals with cantilevers made from monocrystalline silicon by processes of microtechnology. The cantilevers are passive structures and have no transducers. The application as a material measure for the inspection of stylus forces is in the center of investigations. A simple method is the measurement of the deflection of the cantilever at the position of load by the force if the stiffness of the cantilever at this position is known. Measurements of force–deflection characteristics are described and discussed in context with the classical theory of elastic bending. The methods of determining the stiffness are discussed together with results. Finally, other methods based on tactile measurements along the cantilever are described and tested. The paper discusses comprehensively the properties of concrete silicon chips with cantilevers to underpin its applicability in industrial metrology. The progress consists of the estimation of the accuracy of the proposed method of stylus force measurement and the extraction of information from a tactile measured profile along the silicon cantilever. Furthermore, improvements are proposed for approaches to an ideal cantilever.

## 1. Introduction

Monocrystalline silicon, developed for microelectronics, has properties which are excellently suitable for material measures of dimensions and forces [1,2,3,4,5]:The structural perfection;The small thermal expansion;The large thermal conductivity (comparable to those of metals);The elastic constants, only slightly depending on temperature;The considerable hardness; andThe corrosion resistance.

These properties guarantee a high thermomechanical stability. Moreover, the available processes of silicon-microtechnology permit the structuring with maximum precision [3,5]. Nowadays, well-developed silicon bulk micromachining [6] based on wet- [7] or dry-etching techniques [8,9] has enabled the routine fabrication of silicon microstructures with an aspect ratio up to 25 or even higher [10].

Tactile stylus profilometers [11,12,13,14], including long-range scanning probe microscopes [15], have been widely employed in industrial and scientific areas for the characterization of engineering surfaces at micro/nano scales. It is already known that the measurement deviation [16,17], fidelity [18] and even tip wear [19] of tactile surface measurements are strongly affected by the probing force applied in the measurements. Further improvement of the measurement uncertainty of tactile profilometry demands therefore quantitative characterization of the probing force and also the stiffness of various tactile sensors [20,21].

Owing to its excellent mechanical characteristics mentioned above, silicon cantilever-based micro-force sensors [22,23,24,25] have long been developed and employed for probing force measurement. Most of them, however, are integrated or combined with different sensors for force measurement, which result in a relatively higher cost, lower environmental stability and poorer usability of these force sensors, especially for in situ applications.

The paper reports a kind of practical silicon chips with passive cantilevers for the measurement of forces of tactile instruments [26]. Within the chip, a cantilever is situated together with grooves as marks facilitating a scan along the central line of the cantilever and signaling the start and end positions of the tip. The force can be directly determined from the bending of the cantilever without additional transducer functions, e.g., a balance, giving an edge in industrial application.

The cantilever chips are produced by microtechnologies and structured by orientation-dependent etching in aqueous KOH solution. With the crystallographic <100> orientation of the axis, the cantilevers receive a rectangular cross-section [5]. The cantilevers are suitable for certifying.

The bending of the cantilever largely corresponds with the theoretical relations of the classical theory of elasticity [27]. The condition of small bends is fulfilled (the shortening of the length of the bending line projected onto the horizontal baseline can be neglected) [28]. For the stiffness *c* (ratio of the force *F* and appropriate bending *z* at the position of load) of a rectangular cantilever with the length *l*, the width *b* and the thickness *h*, the following expression is valid:(1)$$c=\frac{F}{z}=\frac{3EI}{l^{3}}=\frac{Ebh^{3}}{4l^{3}},$$

*E* means the Young’s modulus of the material (silicon: *E*_<100>_ = 130 GPa), *F* means the working force and *z* means the resulting beam deflection. *I* is called the moment of inertia of area:*I* = *b* · *h*^3^/12.(2)

It will be shown that a value of Young’s modulus *E* can be extracted from measured values of stiffness *c* and the geometrical dimensions *l*, *b* and *h* of the cantilever. The result can be compared with the nominal (official quoted) value. In the case of an approximative agreement, the working force can be estimated using the nominal value of *E*, the geometrical dimensions of the cantilever and the resulting bending *z*. This means that the cantilever represents a material measure for the force which can be traced to a calibrated compensation balance via the cantilever stiffness and its Young’s modulus.

The knowledge of the force working on a surface is an important requirement when using tactile methods of surface metrology (stylus profilometer, AFM) in order to prevent damage to the surface [29,30]. The pressure exerted on the surface by the stylus tip of a profilometer must be large enough for a firm contact but small enough to prevent surface damages. The radius of the stylus tip and the hardness of the surface are determining features. Standardized values for the tip radius are 2, 5, and 10 μm, but other values are also used. The standardized values for the maximum load corresponding to the above-mentioned radii are 0.7, 4, and 16 mN, according to [31]. The tip radius, the force, the distance of measuring points and the surface features must be compatible to make a point load. In each case, the working force should be known. For the measurement of stylus forces, different procedures are well-established [32,33].

The determination of the stylus force of a mechanical profilometer using a profile scanned along a cantilever is a practical method that is simple and feasible at any time. The novelties in the presented paper are detailed investigations of important conditions for the accuracy of this method: The horizontal adjustment of the measured profiles over the cantilever with load and without load;The correction of the bending line under load by the zero-load bending;The analysis of the bending line of the cantilever under the load of the stylus of a mechanical profilometer together with a comparison with the theory of elastic bending;The estimation of the uncertainty of the method; andThe testing of the difference method for determination of the bending force from two points of the bending line.

At first, the dimensions of the cantilevers and the effective Young’s moduli are measured as a basis for the evaluation.

## 2. Materials and Methods

### 2.1. Description of the Cantilever Chips

Figure 1 shows the cantilever chip. The size of the chip is 15 mm × 15 mm with a thickness of 0.525 mm. The chips are manufactured by a two-step etching process in aqueous KOH solution with an etch mask on the front and back side of a silicon {100} wafer. The longitudinal direction of the cantilever is parallel to the crystallographic <100> direction on the wafer [5]. In addition to the shape of the cantilever, a set of additional grooves are etched into the surface, as seen in Figure 1b. They are used for

The longitudinal alignment of the scan (parallel to the grooves between the marks **P_2_**, **P_1_**, und **S**) for the movement of the stylus tip in the median line of the cantilever to avoid a torsion;The horizontal alignment of the profile in the region in front of the cantilever (plane regions between marks **P_2_**, **P_1_**, und **S**);The positioning of the stylus tip on the cantilever between the marks **S** und **E** (as distinct from Young’s modulus *E*);The positioning of load at the end mark **E** for the measurement of stiffness or of force. The end mark **E** consists of an array of etched quadratic hollow pyramids with <110> orientation to ensure an etch stop [5].

The use of microtechnologies ensures a maximum accuracy of the geometrical dimensions of the structures which are inspected by microscopic measurement. The array of grooves **E** near the end of the cantilever marks a defined position of load in the case of bending experiments and is directly depicted in the scanned profile. On the backside, the cantilevers have no boss and can be bent by more than 300 µm downwards (i.e., the wafer thickness minus the thickness of the free-standing cantilever beam).

### 2.2. Experimental Methods

#### 2.2.1. Measurement of the Geometrical Dimensions of the Silicon Cantilevers

The thickness *h* of the cantilevers was measured by a length gauge (Heidenhain-Certo) with the accuracy of ±0.1 µm. From each of the 13 measurements evenly distributed along the length of a cantilever, a mean value was calculated. The mean value of the thickness was between 30 and 200 µm for the different cantilevers. The associated standard deviations were between 0.2 and 0.9%.

A microscope (Vision engineering) equipped with a stage with a digital position display (accuracy of 2 µm) was used to measure the widths and lengths of the cantilever.

The widths were measured at 3 positions (beginning, middle, end) of the cantilever, each at the front and the back. The standard deviations amounted as well to 0.2 to 0.9% so that a rectangular cross-section can be assumed. The lengths of the cantilevers were measured at the left and the right sides. Deviations from the mean value of <0.1 to 0.6% were observed. At the inspection of the front and the back side of the cantilever, its real beginning has to be checked.

#### 2.2.2. Measurement of the Bending Stiffness (Force–Bending Characteristics)

For the measurement of force–bending characteristics by the authors of SiMETRICS, the instrument 1 with the schematic drawing shown in Figure 2 was utilized. It consists of an electronic weighing balance with a computer interface (SAG 285 of Mettler-Toledo GmbH, Gießen, Germany, resolution of 0.1 mg), a stable stand (a ground plate and solid column with a crossbar), three positioning stages for the *x*-, *y*- and *z*-direction mounted on the crossbar, and a rod with a stylus. The stages for *x*- and *z*-directions are equipped with DC motors (Physik-Instrumente GmbH, Karlsruhe, Germany: M-126.DG) and are controlled by a computer. The accuracy of the positioning is 0.1 µm. The *y*-position must be manually adjusted. Another similar instrument 2 situated at the PTB Braunschweig mainly differs by the stage in the *z*-direction using a piezoelectric drive (Physik-Instrumente GmbH, Karlsruhe, Germany: P-725.xDD PIFOC High Dynamics Piezo Scanner).

The cantilever chip was fixed on a glass plate with a thin film of adhesive and placed on a plane silicon plate on the balance. To realize a definite position for loading the cantilever, the stylus tip and the end mark of the cantilever can be monitored with a zoom microscope with a video camera. A housing screens the instrument from air turbulence and temperature fluctuations.

With software, the steps and the loading and unloading speed can be adjusted. The bending *z* and the load of the balance *F* were read out and stored.

At implemented hold points, the stability of the system can be checked. The beginning of the measurement was supported by an automatic search of the moment when the stylus tip comes into contact with the cantilever (“find probe”: a minimum threshold of force is reached; *z* = 0). The input of a “Stop position” or a “Stop force” assigned the end of a measurement. All measurements were performed with increasing (“loading”) and decreasing (“unloading”) bending. The bending stiffness was determined from the force–bending curve with the calculation of a trendline (linear regression); see Figure 3.

A loading/unloading speed of 2 µm/s and bendings up to 100 µm in steps of 2 µm were chosen. Other speeds (e.g., 0.1 µm/s and 5 µm/s) and maximal bendings (e.g., 50 µm and 150 µm) lead to the same value of stiffness, as would be expected.

From the stiffness, an effective value of Young’s modulus *E* can be extracted after Equation (1). It is not the intention of the presented paper to improve the measurement of *E*, but the measured value of *E* is required for the interpretation of the tactile measured profile over the cantilever.

#### 2.2.3. Determination of the Young’s Modulus by the Difference Method

To avoid the propagation of the uncertainty of the exact load position, the so-called difference method was used. In this process, two measurements must be carried out before and after the load position is changed by a precisely defined distance. This can be realized with the motorized positioning table in the *x*-direction. The length of the cantilever and the absolute position of loading are eliminated:(3a)$$c_{1}=\frac{3EI}{l_{1}{}^{3}}$$
(3b)$$c_{2}=\frac{3EI}{l_{2}{}^{3}}$$
(3c)$$E=\frac{1}{3I}\frac{{\left(l_{1}-l_{2}\right)}^{3}}{{\left(\frac{1}{c_{1}{}^{1/3}}-\frac{1}{c_{2}{}^{1/3}}\right)}^{3}}$$

The difference method was applied for three cantilevers. In doing so, the first load point *l*_1_ was chosen near to the end mark **E** and the second load point *l*_2_ was set by the *x*-stage. The experiment has been performed two times with the distance of the load points Δ*x* = (*l*_1_ − *l*_2_) = 1000 µm and Δ*x* = 2000 µm (shifting the *x*-stage by 1000 µm again).

#### 2.2.4. Tactile Measurement of a Profile along a Cantilever

The stylus force of a stylus profiler scanning along a cantilever causes a bending which increases in direction to the free end. This bending can be directly seen in the measured profile *z*(*x*). The cantilevers described in Section 2.1 have an etched groove as end mark E, which can be seen in the profile, too. Knowing the stiffness *c* of the cantilever at this position (immediately before **E**), the force *F* can be calculated by Equation (1) using the bending *z* = *z_E_* immediately before **E**.

Using this procedure, we have to respect three complications:

The chip with the cantilever put on a rest can be inclined against the guide mechanism of the stylus. Thus, the whole profile will be inclined by an angle *φ*. Then, the whole profile must be rotated horizontally with the angle −*φ* and displaced by *z*_zero_ up to *z* = 0 between **P_1_** and **S**, as shown in Figure 4. This adjustment is called “leveling” and must be carefully completed because the end of the long cantilever will be easily displaced.The cantilever can have a bending without an external load $z_{0}$ (“zero-load bending”) because of its own weight and of surface stresses. The bending induced by the own weight can be calculated using a “surface load” $q=F_{w}/l$ where $F_{w}$ is the weight and *l* is the length of cantilever [27]:(4)$$z_{w}=\frac{3ql^{4}}{2Ebh^{3}}=\frac{ql^{4}}{8EI\;}.$$

The values of *z*_w_ are shown in Table 1.

The bending caused by surface stresses cannot be calculated in a simple way. Both components of zero-load bending *z*_0_ cannot be separated and must be measured by a non-tactile technique, e.g., by an optical instrument. In addition, this measurement must consider a potential angle of incline of the chip.

3.The angle of rotation *φ* and the shift *z*_zero_ have a strong influence on parameters extracted from a measured profile. The determination and compensation of *φ* and *z*_zero_ have to be calculated very carefully. Three methods based on different regions of leveling (ROL) are possible, as shown in Figure 5:
i.Single-end approach (SEL): leveling the cantilever beam using the region between the marks **P_1_** and **S** as ROL (Figure 1b), i.e., the red line along the central axis of the cantilever beam, as shown in Figure 5;ii.Dual-end approach (DEL): using two narrow regions as ROL: close to the mark S and diagonal to the frame, i.e., the red and blue line along the central axis of the cantilever beam (the tactile measurement cannot use this method);iii.Chip leveling approach (CL): using the long and narrow regions on the chip and parallel to the cantilever axis as ROL, i.e., the two green lines in Figure 5.

**Figure 5 sensors-22-06253-f005:**
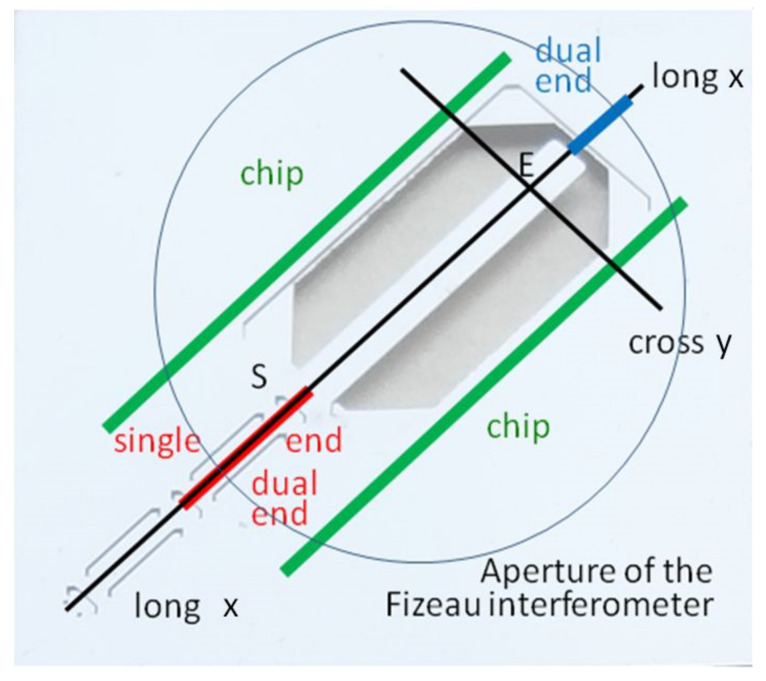
Schemes of the leveling approaches for evaluation of the zero-load bending of cantilevers and of the autofocus measurement of profiles for the determination of the zero-load bending (black lines). The direction “cross” yields *z*_0*E*_, while the longitudinal direction “long” yields the profile *z*_0_(*x*).

The angle of inclination *φ* of these ROL can be determined by the least-square linear fitting (trendline) *z* = *x* tan *φ* + *z*_0_. Then, the rotated profile has the coordinates *x*′, *z*′:(5a)$$x^{\prime }=x\cdot \cos\varphi +z\cdot \sin\varphi \;\;\;\;\;\;\;\;\;\;\;\;\;\;\;\;\;\;\;\;\;\;z^{\prime }=-x\cdot \sin\varphi +z\cdot \cos\varphi$$

Normally, the angle *φ* is very small, and we can assume cos *φ* ≈ 1 with the result that
(5b)$$\frac{x^{\prime }}{\cos\varphi }=x^{\prime }=x+z\cdot \tan\varphi \;\;\;\;\;\;\;\;\;\frac{z\prime }{\cos\varphi }=z\prime =-x\cdot \tan\varphi +z$$

In the case of a profile with the length x ≫z and where tan *φ* is small, the coordinates of the rotated profile can be calculated by:(5c)$$x^{\prime }=x\;\;\;\;\;\;\;\;z\prime =-x\cdot \tan\varphi +z$$
where tan *φ* is the slope of the trendline (linear regression). The simplified calculation of the new coordinates of the profile can be called leveling (instead of rotation).

The elastic bending caused by the stylus force is the result of the difference of the bending *z*(**E**) = *z***_E_** extracted from the profile, as shown in Figure 4, and the zero-load bending *z*_0**E**_: *z***_E_** − *z*_0**E**_. Consequently, two parameters must be known for the determination of the stylus force: the stiffness *c* at **E** and the zero-load bending *z*_0**E**_ at **E**.

Then, the stylus force *F* can be obtained by Equation (6)
(6)$$F=\left(z_{E}-z_{0E}\right)\cdot c.$$

#### 2.2.5. Measurement of the “Zero-Load Bending”

The zero-load bending can be obtained by an optical measurement. Two instruments were used: an autofocus sensor and a Fizeau interferometer.

Measurement with the autofocus sensor

With the autofocus sensor (AF 16 with stage Hyperion of OPM, Ettlingen, Germany), we have measured the zero-load bending *z*_0*E*_ at the end mark **E** from a profile in front of and parallel to the grooves of the end mark and perpendicular to the cantilever, and we compare the direction “cross” in Figure 5. Alternatively, a profile *z*_0_(*x*) along the cantilever can be measured, which is called “long” in Figure 5. So, the resulting profile must be leveled, too. The horizontal alignment can be realized with the single-end method. The marks **P_1_**, **S** and **E** are shown in the profile “long”, and the zero-load bending *z*_0**E**_(long) can be directly read out before the end mark **E**. Nevertheless, some uncertainties arise. *z*_0**E**_(cross) measured along “cross” at the end mark perpendicular to the cantilever can deviate from *z*_0**E**_(long) if the chip has a curvature (“bow”, see Section 3.4.1). The resolution of this instrument amounts to 10 nm in the *z*-direction and 0.5 µm in the *x*- and *y*-direction.

Measurement with the Fizeau interferometer

A Fizeau interferometer for topography measurements of nearly flat workpieces [34] is used to measure the 3D topography of the silicon micro-force standards under zero load. The measurement field of this interferometer amounts to 10 mm in diameter, with a lateral resolution of 17 µm and an uncertainty of λ/20 for flatness measurements at a wavelength of λ = 633 nm. From the 3D topography, 2D-line profiles can be extracted.

#### 2.2.6. Evaluation of the Whole Bending Line

A measured profile consists of a series of pairs of coordinates *z*(*x*) indicating the bending *z* at the position *x* caused by the constant force *F* acting at the position *x*. In theory, the profile *z*(*x*) is described by a third-order parabola:(7)$$z=\frac{F}{3EI}\cdot x^{3}=\frac{4F}{Ebh^{3}}\cdot x^{3}.\;$$

A plot of the bending *z* against *x*^3^ for the cantilever yields to a linear function with the slope *F*/(3*EI*). From this slope, the force *F* (if Young’s modulus *E* is known) or the modulus *E* (if *F* is known) can be calculated, provided that the width *b* and the thickness *h* are known. In doing so, the fact is neglected that the weight of the last part of the cantilever after the end mark **E** slightly increases the working force. Furthermore, the zero-load profile must be subtracted from the tactile measured profile or it can be neglected if it is sufficiently small.

Finally, the difference method can be applied to the points of a measured profile after subtracting the zero-load profile, too. Series of pairs of points in a distance Δ*x* must be selected. Using the nominal value of Young’s modulus *E* of the cantilever, the stylus force results from each pair of points (analogous to Equation (3c)):(8)$$F=\frac{3EI\cdot {\left(\sqrt[{3}]{\left(z\left(x\right)\right)}-\sqrt[{3}]{\left(z\left(x-\Delta x\right)\right)}\right)}^{3}}{{\left(\Delta x\right)}^{3}}.$$

The function *F*(*x*, *z*) should have an approximately constant value, and a mean value can be calculated.

## 3. Results

### 3.1. Summary of the Measured Dimensions of the Used Cantilevers

Table 2 shows the mean values of the dimensions of the used cantilevers measured under conditions described in Section 2.2.1.

### 3.2. Estimation of Young’s Modulus from the Measured Stiffness of the Cantilever

Bending a silicon cantilever with the longitudinal axis <100> by a perpendicular force *F*, the Young’s modulus with a nominal value of *E*_<100>_ = 130 GPa takes effect. In the case of small elongations, the classic theory of elastic bending yields that the stiffness will follow Equation (1). By anisotropic etching in aqueous KOH-solution of the <100>-cantilevers, a rectangular cross-section arises [5], and the moment of inertia of area follows Equation (2). Using the values of width *b*, thickness *h* and length *l* which are individually measured on each cantilever, values for *E* can be calculated; see Table 3. *l* must be corrected if the load position differs by a distance *ξ* from the end mark **E**. Most measurements of stiffness were made by co-workers at SiMetricS; some cantilevers were measured at the PTB.

### 3.3. Results of the Difference Method

At each point of load, five series of force–bending characteristics were measured, each with increasing and decreasing load. From the mean value of the stiffnesses, the Young’s modulus was calculated using Equation (3c). If the measurement series were made at an initial load position *x*_A_ followed by a load position *x*_A_-1000 µm, and immediately after it at a load position *x*_A_-2000 µm (repeated shift by 1000 µm), the second and the third series can be analyzed with Δ*x* = (*l*_1_ − *l*_2_) = 1000 µm. The results are collected in Table 4.

It must be noticed that very few measurements result in values of *E* that deviate by more than 10% from the nominal value (not included in Table 4). These measurements have produced outliers. Reasons could be imprecise positioning of the tip, unperceived slipping of the chip, or instabilities of the bearing (dust). The chip 200-3 was not measured with Δ*x* = 2000 µm (because of a defect).

### 3.4. The Optical Measurement of the Zero-Load Bending

#### 3.4.1. Measurement of Line Profiles with the Autofocus Sensor

Profiles in the direction “long” (Figure 5) were only measured on chips with nominal thicknesses *h* = 45 µm (45-1) and *h* = 70 µm (70-4). Figure 6a shows these profiles leveled by the single-end approach. It can be seen that the chips have a curvature presumably resulting from the “bow” of the Si-wafer [5], p. 5. Determining the zero-load bending *z_E_*_0_ as the height over the zero level of the single-end leveling gives larger values; then, they are determined from the dual end leveled profiles in direction “cross” at the *x*-position of the end mark **E** (Figure 6b and arrows in Figure 6a). Excluding the region of the cantilever from the profile, the region of the chip can be fitted, e.g., by a 2nd power polynomial (chosen due to simplicity).

The mean values resulting from the middle parts of the dual end leveled cross-scans in Figure 6b are listed in Table 5. The high standard deviation for the thickness *h* = 200 is induced by the noise of the autofocus sensor. The chip regions beside the cantilever show a symmetrical slope. Fitting these regions of the profile by a 2nd power polynomial too (without cantilever) shows the bow of the chip in the cross-direction. The negative z-values of the trendline below the cantilevers have to be added to the *z*_0*E*_ of the scan as an offset.

Because the tactile profile cannot cross the end of the cantilever, only the single-end leveling is possible. Therefore, the zero load profile in the direction long must also be single-end leveled to match both profiles. If the profile is due to mathematical analysis, a careful single-end leveling with ROL between **S** and **B** (**B**: beginning of the cantilever) must be realized.

The scattering of the points of the zero-load profiles can be explained by an unsteady environment of the instrument and by small deviations of the thickness of the cantilever resulting from the etch process.

#### 3.4.2. Measurement of Zero-Load Bending with the Fizeau Interferometer

For demonstration, the measurement of zero-load bending with the Fizeau interferometer should be described. Figure 7a shows the topography of the microforce standard FC30 21-4 with a cantilever thickness of 30 µm (nominal value) from top view. The measured 3D topography corrected for the tilt is illustrated in Figure 7b. The line profiles measured on the cantilever at *y* = 0 mm and those on the adjacent substrate at *y* = 0.8 mm are depicted in Figure 7c.

The zero-load bending of four microforce standards (cantilevers) with a nominal beam thickness of 30 µm has been experimentally determined. Because of the limited aperture of the interferometer, the cantilever was displaced by 1.2 mm in the *x*-direction for a longer ROL with the single-end approach (the cantilever was measured in two measurements with a displacement of 1.2 mm in x. Then, the single measurements were stitched together to form one topography). The results are summarized in Table 6.

The small *z*-values of the zero-load profile show a considerable scattering. A polynomial fitting is advantageous for the subtraction from the points of the bending line. A good approximation is achieved by a polynomial of fourth power [35,36].

### 3.5. Evaluation of a Tactile Measured Profile along the Cantilever

#### 3.5.1. Determination of the Stylus Force from the Bending at the End Mark **E**

The procedure will be demonstrated using the profiles of the two cantilevers 45-1 (*h* = 45 µm) and 70-4 (*h* = 70 µm), as shown in Figure 4. The profiles of both cantilevers were measured using the same tactile profiler Dektak XT (Bruker, Tucson, AZ USA) without change of the stylus force. These profiles can be measured only up to the end of the cantilever. Consequently, the single-end leveling is possible only: rotating the profiles horizontally and zero setting the ROL (subtraction of a constant level). After that, it must be checked whether the profile of the cantilever starts (distance from **S** about 600 µm) at z_zero_ = 0 and nearly trends horizontally. If not (in excess of the scattering of points), a correction is recommended. After these procedures, the bending of the last five points before the end mark **E** were read out for calculation of a mean value *z***_E_**, as shown in Figure 8 (more points cannot be used because of the slope of the bending line). Analogously, the zero-load bending *z*_0**E**_ was determined from the zero-load profile measured optically with the autofocus instrument OPM.

The bending *z***_E_** contains the zero-load bending *z*_0**E**_ which must be subtracted, as shown in Equation (6). For exact measurements or calibrations, the zero-load bending *z*_0**E**_ must be measured for an individual cantilever. Knowing the stiffnesses (Table 3) of the used cantilevers, the stylus force can be calculated by Equation (6). The resulting values of the stylus force are shown in Table 7. An agreement of both values can be seen in view of the calculated uncertainties, as shown in Table 3.

#### 3.5.2. Determination of the Stylus Force Using the Whole Profile

At first, the part of the free bending cantilever must be cut out from the horizontally leveled profile. In reference to the end mark **E** (Figure 9), the beginning of the cantilever is located at the distance *l* (free bending length; see Table 2). As examples, the profiles over the cantilevers 45-1 and 70-4 are considered. In the measured profiles, the end marks **E** are at the positions *x* = 12075 µm (45-1) and *x* = 12085 µm (70-4); see Figure 9. With *l* = 6013 µm resp. *l* = 5946 µm, the free bending length begins at *x*_beginning_ = (12075 − 6013) = 6062 µm resp. *x*_beginning_ = (12085 − 5946) = 6139; see arrows **B** in Figure 9.

From the measured profile, all points with *x* < 6060 resp. 6140 µm and *x* > 12075 resp. 12085 µm must be cut out. While the procedure described in 3.5.1 uses only two points, *z*_0_ and *z*_0**E**_, now, all points of the bending profile are required, i.e., we have to check the shape of the bending line: the beginning of the profile must have *z* = 0 and d*z*/d*x* = 0. If it is not so, the leveling of the bending line has to be improved with the beginning of the profile as ROL (from **B** up to ca. 300… 400 µm). The same holds true for the zero-load bending line.

Now, the zero-load profile of the cantilever must be subtracted from the tactile measured profile. By this process, the “bow” is corrected, too, provided the tactile profile has the same “bow”. Figure 10 shows the bending profiles *z*(*x*) for the cantilevers 45-1 and 70-4 as measured and definitively single-end leveled and after subtraction of the zero-load profile (also definitively single-end leveled).

A plot of all (*z* − *z*_0_) values against *x*^3^ should generate a linear function. From the trendline (linear regression), a slope of (*z* − *z_0_)/x*^3^ results, as shown in Figure 11.

With Equation (1), the slope is related to *F*:(9)$$F=3EI\left(\frac{z-z_{0}}{x^{3}}\right)$$

Using the units m for *x* and *z* as for all cantilever dimensions and Pa for Young’s modulus *E,* the force *F* is given in N. For the comparability with the results in Table 7, the measured Young’s modulus *E* (Table 3) must be used.

The results are summarized in Table 8.

The results deviate by 1% from the determined stylus forces even with the results described in Section 3.5.1; see Table 7.

The use of the difference method for the determination of the stylus force yields partially reasonable results; see Figure 12. Δ*x* = 500 µm, Δ*x* = 1000 µm and Δ*x* = 2000 µm were tested.

Because the first point of the first pair cannot lie before the beginning of the cantilever, the first value of *F* can be determined for the position *x* = Δ*x* on the cantilever. At the beginning of the cantilever, the very small *z* values produce improper results. Additionally, there is no proportionality of *z* with *x*³ in this region because of the slightly increasing thickness of the cantilever.

The influence of the displacement Δ*x* on the calculated values of *F* is small for *x* >> Δ*x*, as can be seen in Figure 12. The scattering of the values is increased with a decrease in Δ*x*. The mean value of *F* is calculated over the range of 1000 µm beginning from the end in direction to the beginning of the cantilever (200 values). In the case of cantilever 45-1, the use of Δ*x* = 500 µm produces results in this end region in good agreement with the other described methods; see Table 9. Against it, the results of cantilever 70-4 deviate clearly from the values determined by the other methods. The reason can be the uncertainty of the leveling procedure in the case of very small z-values of the profile with large scattering.

We can conclude: using the silicon cantilevers, the determination of the Young’s modulus *E* (with known stylus force or cantilever stiffness) or of the stylus force *F* (with known cantilever stiffness or Young’s modulus *E*) is possible.

## 4. Discussion

### 4.1. Estimation of the Young’s Modulus from the Measured Stiffnesses of the Cantilever

Other than some outliers, the determined values of the Young’s modulus for the monocrystalline silicon cantilevers lie between 128 and 133 GPa, corresponding to an interval of ±2%. Generally, surface tensions causing the zero-load bending will influence the measured Young’s modulus, too, in particular if the thickness of the cantilever is small. Other sources of the deviations are the uncertainties of the thickness and the effective length (depending on the load position) of the cantilever influencing the results by its 3rd power. Measuring errors or fluctuations of the thickness has a particularly large influence when the thickness is small such as with the cantilevers with the nominal thickness of 45 µm; see Table 2 and Table 3. The tactile measurement of the thickness cannot record the region of 0.4 mm at the beginning of the cantilever because the probe diameter cannot influence the values in Table 2 as a consequence. An increased thickness at the beginning of the cantilever can arise from the etching process (“hollow mirror effect” [5], p. 87); see Figure 13. This effect can be avoided by a modified technology using a compound of two wafers: a thin wafer with the thickness of the cantilever bonded on a stable wafer at the bottom. Then, from the bottom, the region of the cantilever is etched up to the bond oxide followed by etching the shape of the cantilever from the top, removing the oxide at the end of the process. A cantilever with uniform thickness results.

These effects explain a contribution to the larger values of Young’s modulus calculated from the stiffness of the cantilevers with the thickness of 45 µm; see Table 3.

The backside of the cantilevers corresponds to the “etch ground” having a surface quality that decreases with increasing depth [5]. The largest depth must be etched for the cantilevers with *h* ≈ 30 µm and 45 µm.

Uncertainties of the length *l* result less from the measurement of the length but from uncertainties of the adjustment of the load point. The end mark **E** has a total extent of about 60 µm in the direction of *l*. Consequently, the load point must be exactly defined in relation to **E**. This is difficult in the case of loading by a tip with a large radius.

The latter problem can be avoided by using the difference method. However, the thereby obtained values of the Young’s modulus have nearly the same scattering even in the case of measurements of the same cantilever with constant geometrical dimensions. The maximum uncertainty of the stage positioning is 1 µm, which is alone not enough to explain the error. The above-mentioned instabilities of the bearing (dust) can be another reason.

### 4.2. Measurement of the Stylus Force Using a Profile along the Cantilever

#### 4.2.1. Determination of the Force from the Stiffness and the Bending at the End Mark **E**

This is the simplest method to determine the stylus force. Using the geometrical dimensions of the cantilever and the nominal value of Young’s modulus (130 GPa), the stiffness can be calculated, and the bending *z***_E_** at the end mark **E** can be read out directly from the scanned profile. After subtraction of the zero-load bending *z*_0**E**_, the force can be calculated by Equation (6).

Having measured and certified values of stiffness *c* and zero-load bending *z*_0**E**_, an analogous procedure can yield a stylus force with preferable accuracy.

The uncertainty of the stylus force determined by this method results from

The uncertainty of the stiffness *δc* (given by the certification);The uncertainty of the bending *δz***_E_** at the end mark (must be found by the user);The zero-load bending *δz*_0**E**_ at the end mark (given by the certification).

The bending *z***_E_** at the end mark must be extracted by the user from the tactile profile. The profile has to be leveled, as shown in Figure 4. The value *z***_E_** depends on the uncertainty of the leveling slope *S*_ROL_ (tan *φ*) and on the *z*_zero_ at the beginning of the cantilever. The user has to ensure small values of the *z*_zero_ and of *S*_ROL_ (i.e., the zero level and the parallelism of the guide of the stylus and the cantilever chip). The uncertainties $\delta z_{E}$ or $\delta z_{0E}$ can be simply calculated as follows:(10)$$\delta z_{E;0E}{}^{2}=\delta z_{\mathrm{tac}/\mathrm{AF}}{}^{2}+\delta z_{\mathrm{ROL}}{}^{2}+\delta z_{lat}{}^{2}$$

For the measurements with the tactile profilometer (Dektak) or with the autofocus sensor (AF 16), $z_{\mathrm{tac}}$ = 1 nm resp. $z_{\mathrm{AF}}$ = 10 nm are used as examples. The uncertainty of the profile caused by the leveling process is given by the slope *S*_ROL_ and the zero offset *z*_zero_: $\delta z_{\mathrm{ROL}}=L\cdot \delta S_{\mathrm{ROL}}+\delta z_{zero}$ (*L*: length from the origin to **E** of the leveled profile; *δz*_zero_: the standard deviation of the *z* values inside ROL after leveling and displacement to *z* = 0). The uncertainty contribution of the profile caused by the uncertainty of the lateral position *δx* is described by $\delta z_{\mathrm{lat}}=z^{\prime }\left(x_{E}\right)\cdot \delta x={\left(\frac{\Delta z}{\Delta x}\right)}_{E}\cdot \delta x$. The slope ${\left(\frac{\Delta z}{\Delta x}\right)}_{E}$ is calculated as the mean value over the last five measuring points before the end mark **E**; see Figure 8. δx is the uncertainty to meet the end mark **E** (the half diagonal of the little squares; see Figure 1a, plus the distance of the measuring points): δx≈12 µm. Table 10 shows the resulting budgets of the uncertainty of the cantilevers 45-1 and 70-4.

Therefore, the uncertainty of the stylus force amounts to 2.6 µN (cantilever 45-1) and 2.7 µN (cantilever 70-4); see Table 7. The stylus forces determined with these cantilevers differ inside these error limits by 1.3%.

To keep the uncertainty small, the measurement of the stylus force should be performed with a cantilever bending at the end mark of more than *z***_E_** = 3 µm by choosing a suitable stiffness and thus thickness.

#### 4.2.2. Determination of the Stylus Force by Evaluating the Whole Profile

The whole profile contains a very large number of points representing bending elongations *z*(*x*) at load points *x* (along *l*). In this manner, the plot of *z*(*x*^3^) has a statistical basis reducing the influence of random errors.

For discussing the calculation of the stylus force based on the whole profile, a revisal of the mathematical description of the bending profile is advantageous. The analysis of measured profiles according to Equations (7)–(9) assumes a function:(11)$$z\left(x\right)=a\cdot x^{n}\;\mathrm{with}\;n=3.$$

To examine the validity of this simple model, the bending profile can be pictured with a double logarithmic scale; see Figure 14.

A clear power function does not appear until *x* = 1000 µm (*l*/6). In the last 2/3 of the zero load corrected bending line, the slope shows values of about 2.9 ≈ 3; see Figure 14. Hence, we can assume the validity of Equation (11) at least in the last 2/3 of the cantilever. Thereby, the determination of the force with Equation (7) as well as the difference method with Equation (8) is justified. The forces resulting from Equations (7) and (9), as shown in Table 8, nearly agree (±1%) with that by using *c* and *z*_0**E**_, see Equation (6) and Table 6, if the measured Young’s modulus of 138.7 resp. 130.7 GPa is used. In the case of cantilever 45-1, the force determined by the difference method with Δ*x* = 500 µm agrees with these values also inside 1%, whereas the values of the cantilever 70-4 differ significantly. Apparently, the difference method is very susceptible to faults of leveling.

The deviation of the bending line from the power function at the beginning of the cantilever can partly result from the increased thickness of the cantilever toward the beginning over a range of the etch depth: therefore, about 300…500 µm [5]; see Figure 13. An increased thickness causes smaller bendings; see Figure 14b. The *z*-values at the beginning of the cantilever will be the result of the roughness and the noise of the measuring instruments as a lower limit of measurable values. This effect disappears for *x* > 1000 µm when the bending exceeds 10 nm, as can be seen in Figure 14. In Figure 11, the power function *z*~*x*^3^ seems to be valid, but the bad points with *x* < 1000 µm lie close to *x* = 0. Even in the bending line *z* (*x* < 1000 µm), the effect cannot be noticed because of the very small values of *z*; see Figure 11.

#### 4.2.3. The Zero-Load Bending of the Cantilevers

The zero-load bending is not dominated by the own weight of the cantilever. The investigated cantilevers show small zero-load bendings in the opposite direction to gravity. The magnitude increases with the decreasing thickness of the cantilever; see Figure 15.

Taking into account the “bow” of the chip, the zero-load bendings *z*_0**E**_ at **E** received from measurements in the directions “cross” and “long” are in good accordance and reveal a clear dependence on the thickness of the silicon cantilevers.

For the analysis of the bending line of the cantilever under zero load in the direction “long”, the profile must be single-end leveled carefully at the base of the ROL between the start mark **S** and the beginning of the cantilever. Then, the profile can be analyzed by a double logarithmic representation analogous to the loaded cantilever (Figure 14) shown in Figure 16.

A simple power function does not arise. The power tends to 2 corresponding with the bending line of a bimetal [27]. The bending lines of the analyzed cantilevers under zero load can be also successfully fitted by polynomials of second or fourth order as proposed by other authors for a cantilever with surface tension [35,36].

The surface tensions can be developed from the different processing of the upper (polished) and lower (etched) surface. Consequently, a more suitable processing technology stands to gain a minimized zero-load bending: a short etch step on the upper surface (etching the wafer before all processes) or the use of a bonded compound of two polished silicon wafers (with a top wafer having the thickness of the aspired cantilever). The last alternative guarantees a better surface quality on the back side, too.

## 5. Conclusions

A silicon cantilever is a convenient tool for the measurement or revisal of the stylus force of tactile stylus profilometers. A simple handling and the absence of electromechanical transducers are advantageous for industrial application.

At the base of the classic theory of elastic bending, the silicon cantilever represents Young’s modulus *E*_<100>_ of silicon and can be used as a material measure for small forces.

From a measured profile *z*(*x*), the stylus force can be extracted by three ways: The stiffness *c* of the cantilever is known for a definite loading position *z***_E_** (at the end mark **E**) together with the bending at this position *z*_0**E**_ without load. From the measured profile that must be single-end leveled and set to zero in the region between **P_1_** and **S**, the bending *z***_E_** can be read out. Then, the stylus force results from Equation (6). As application data, the stiffness *c* and the zero-load bending *z*_0**E**_ can be certified, too. An accuracy of 1% will be reached. The paper proves that this is the most solid method for industrial application.From the whole tactile measured profile, an optically measured profile of the zero-load bending is subtracted. The plot of the difference values of *z* against *x*^3^ results in a linear function. From the slope, the stylus force *F* can be calculated based on a large number of measured points.The so-called “difference method” is also based on a measured profile after zero-load correction. In this process, the force is extracted from the measured points of the end region of the profile. The inaccurate points at the beginning of the cantilever are not used, because the bending is in the order of magnitude of the instrument noise.

Not least, a simple scan along the cantilever allows a quick monitoring of the measuring force of a tactile instrument in industrial application.

The uncertainties in the leveling procedures could be minimized by the use of silicon wafers with larger thickness and better flatness when manufacturing the chips with cantilevers. A short etch step of the wafers before manufacturing the chips results in the same surface processing of the front and back side of the cantilevers. So, the surface stresses as the source of zero-load bending could be minimized. Finally, the use of a wafer compound (SOI-wafer: a stack of a thick wafer for a stable chip — oxide — a thin wafer with the thickness of the cantilever) could yield a cantilever with constant thickness and the same surface processing.

## Figures and Tables

**Figure 1 sensors-22-06253-f001:**
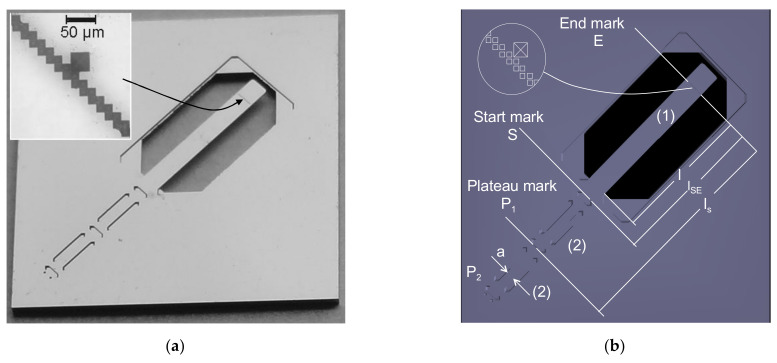
Description of the microforce cantilever chips [26]: (**a**) View on an etched silicon cantilever chip as a force standard (cantilever parallel to <100>-direction) with zoomed view of the end mark **E**; (**b**) Designation of the marks.

**Figure 2 sensors-22-06253-f002:**
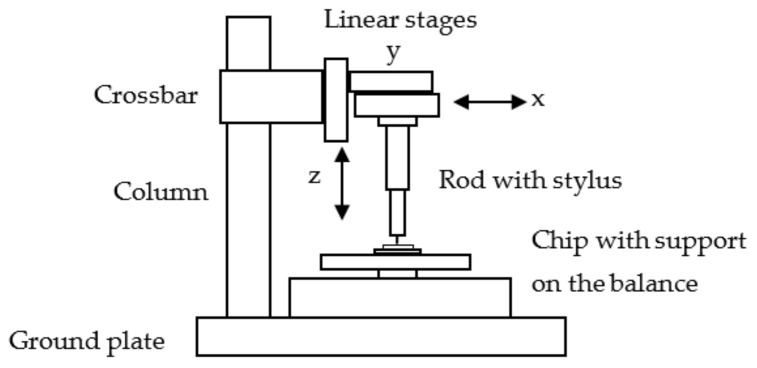
Scheme of instrument 1 for the measurement of the force–bending characteristics.

**Figure 3 sensors-22-06253-f003:**
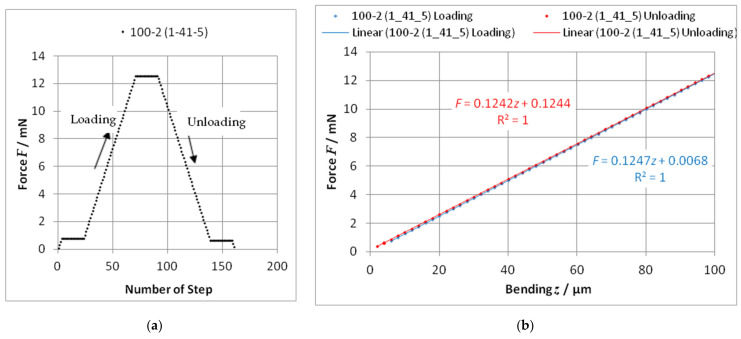
Plot of a force–bending characteristic: (**a**) Survey over the measuring sequence with 2 µm steps (stop points at a small force and at the maximum force); (**b**) Determination of the stiffness *c* = *F*/*z* with linear trendlines. The specification of the cantilever 100-2 is given in Section 3.1; (1_41_5) means the 5th single measurement).

**Figure 4 sensors-22-06253-f004:**
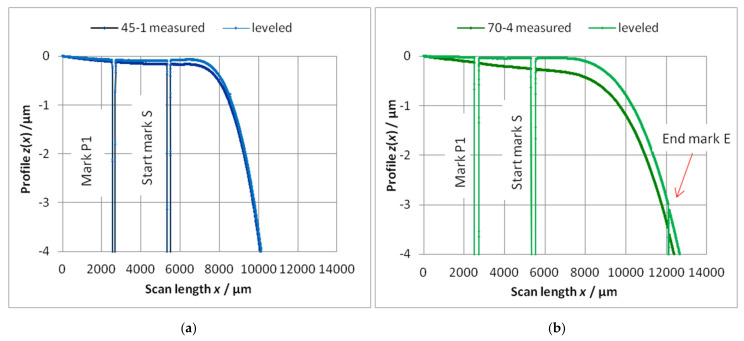
These figures show the profiles measured on the cantilevers (**a**) 45-1 (blue) and (**b**) 70-4 (green). Dark: The profile as measured. Bright: The profile rotated (single-end approach, see Figure 5) by the angle *φ* determined between the marks **P_1_** and **S**. Hereafter, the bending line is slightly displaced by *z*_zero_ to the negative direction by the procedure of rotation. A shift of the profile up to *z*_zero_ = 0 must follow (the combination of the procedures rotation and shift is called “leveling”. Then, the bending line begins at approximately *z* = 0 with a slope of zero. The specification of the cantilevers 45-1 and 70-4 are given in Section 3.1).

**Figure 6 sensors-22-06253-f006:**
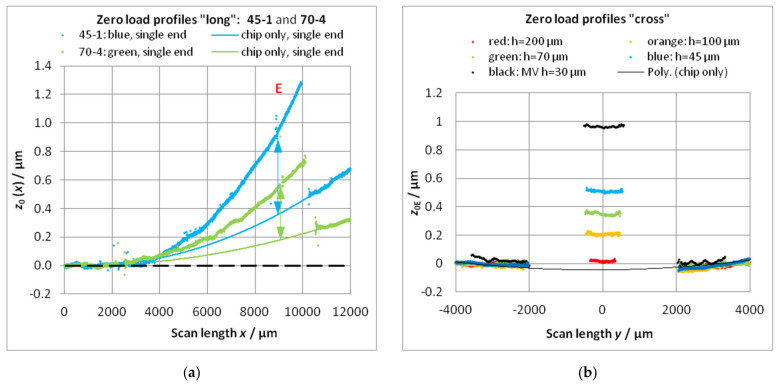
(**a**) The zero-load bending *z*_0_ in the middle of the cantilevers in the direction “long” of the chips 45-1 and 70-4 after single-end leveling. With increasing thickness, the zero-load bending decreases (measured with the autofocus sensor). (**b**) Examples of the zero-load bending *z*_0**E**_ in the middle of the scans in the direction “cross” (after dual end leveling), depending on the cantilever thickness. With increasing thickness, the zero-load bending decreases (black: measured with the Fizeau interferometer; others: measured with the autofocus sensor). *z*_0**E**_—values from the profile “cross” correspond with the arrows in (**a**). In addition, in the cross-direction, the chips can have a little bow (only the largest bow is implemented in (**b**)).

**Figure 7 sensors-22-06253-f007:**
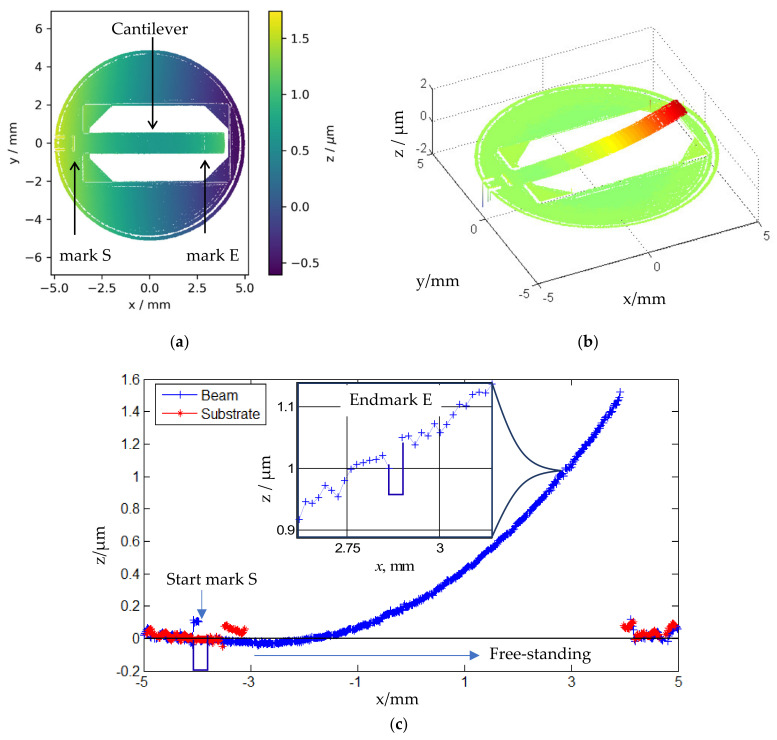
Topography measurement of a silicon microforce standard using a Fizeau interferometer: (**a**) 2D height map of the cantilever-based microforce standard; (**b**) 3D topography of the microforce standard; (**c**) Comparison between line profiles measured on the cantilever and parallel to the beam on the frame (substrate). (Note: Due to the inability of the interferometer to measure discontinuous surfaces, the profiles of step-like marks **E** and **S** cannot be rendered accurately).

**Figure 8 sensors-22-06253-f008:**
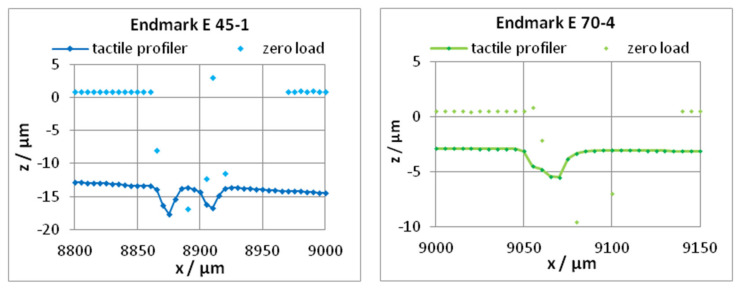
The bending *z***_E_** of cantilevers at the end mark **E** caused by the stylus force of a tactile profiler: magnified region near the end mark **E** (left: *z***_E_** = −13.436 µm, cantilever 45-1, the stylus crosses two grooves of the marker array; right: *z***_E_** = −2.931 µm, cantilever 70-4). Additionally, the optical measured points of the zero-load bending near the end mark are included in the diagram (left: *z*_0**E**_ = 0.938 µm, cantilever 45-1; right: *z*_0**E**_ = 0.502 µm, cantilever 70-4).

**Figure 9 sensors-22-06253-f009:**
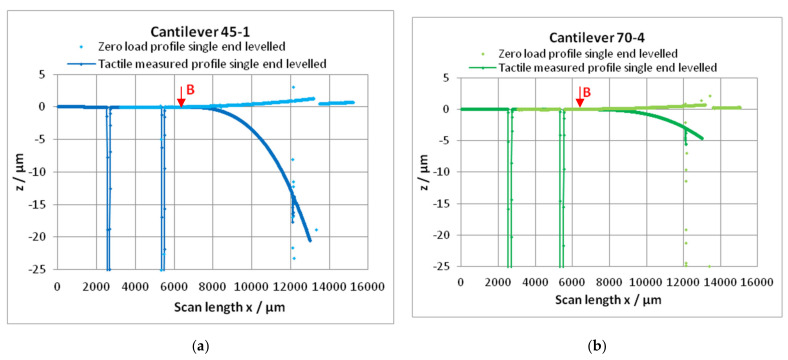
Tactile measured profile (dark) and the zero-load profile (bright, measured with autofocus sensor) (**a**) of the cantilever chip 45-1 (nominal thickness 45 µm) and (**b**) of the cantilever chip 70-4 (nominal thickness 70 µm). **B**: beginning of the cantilever.

**Figure 10 sensors-22-06253-f010:**
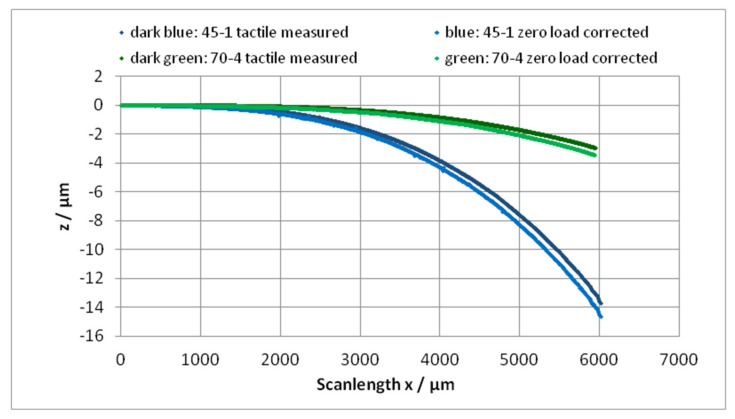
The bending lines of the cantilevers 45-1 and 70-4 before and after zero-load correction.

**Figure 11 sensors-22-06253-f011:**
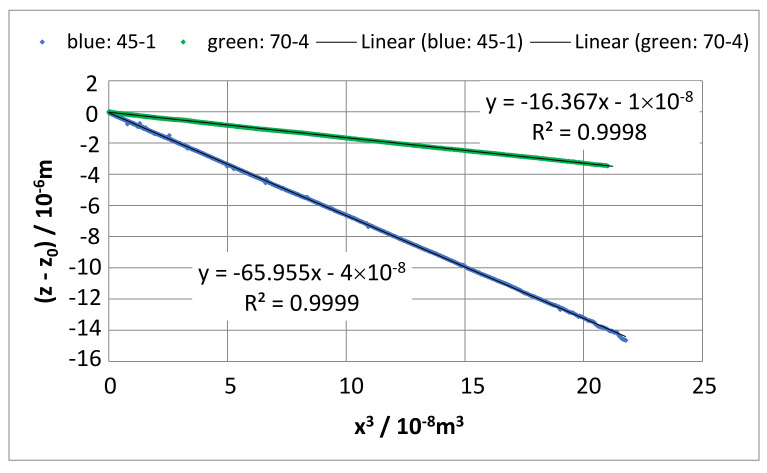
The negative bending elongation of the cantilever as function of the 3rd power of the load position *x*. The trendline (linear regression) gives the slope Δ*z*/Δ(*x*^3^) = *F*/(3*EI*), (Equation (7)).

**Figure 12 sensors-22-06253-f012:**
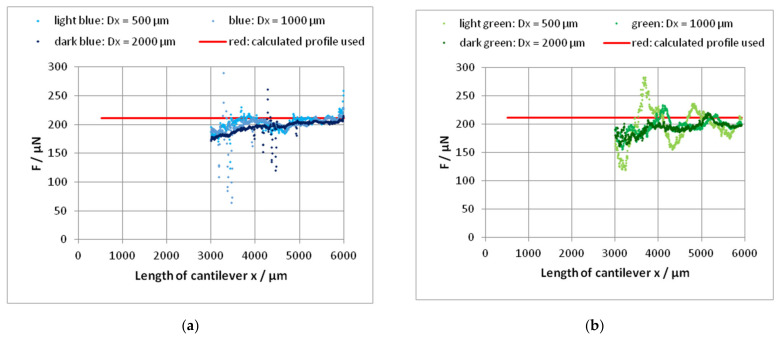
Determination of the stylus force from a tactile measured profile (zero load corrected) of the cantilevers 45-1 (**a**) and 70-4 (**b**) using the difference method after Equation (8); Δ*x*: 500 µm; 1000 µm; 2000 µm. The red lines result from a function *z*(*x*)= *a* · *x*^3^ fitted at the base of the measured values; see Table 8.

**Figure 13 sensors-22-06253-f013:**
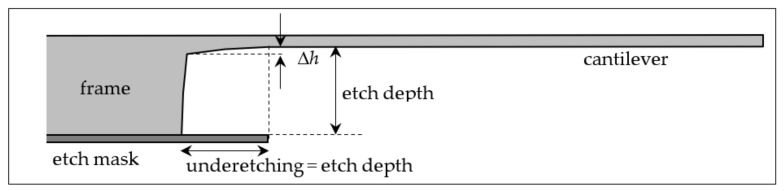
Illustration of the increasing thickness Δ*h* of a cantilever toward the frame under the etch mask. Δ*h* amounts of about 3.5, 4.5, 5 and 5.3 µm for thicknesses of 200, 100, 70 and 45 µm (and respective etch depths of 325, 425, 455 and 480 µm). The length of the underetching is approximately equal to the etch depth from the beginning of the cantilever.

**Figure 14 sensors-22-06253-f014:**
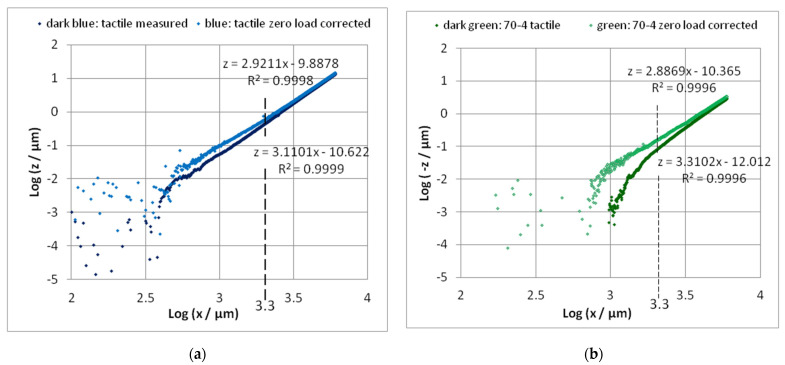
Double logarithmic analysis of the bending line of cantilevers 45-1 (**a**) and 70-4 (**b**). Because the values at the beginning of the cantilevers are strongly scattered (partially negative), the analysis makes sense for x > 2000 µm (Log 2000 = 3.3). Determination of the slope of the linear part (from 2000 µm) reveals the power of the bending line.

**Figure 15 sensors-22-06253-f015:**
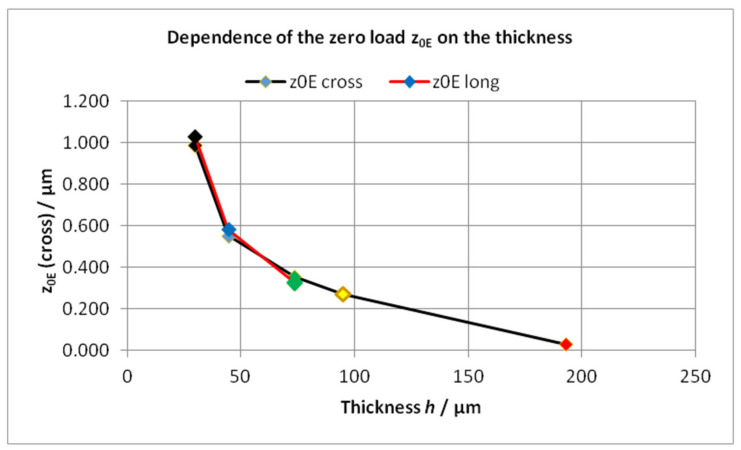
The dependence of the zero load *z*_0**E**_ on the thickness measured in the direction “cross”, (black line, values bow corrected, Table 5) and “long” (red line, single-end leveled and bow corrected). The values for the thicknesses *h* = 30 µm are mean values of 4 cantilevers.

**Figure 16 sensors-22-06253-f016:**
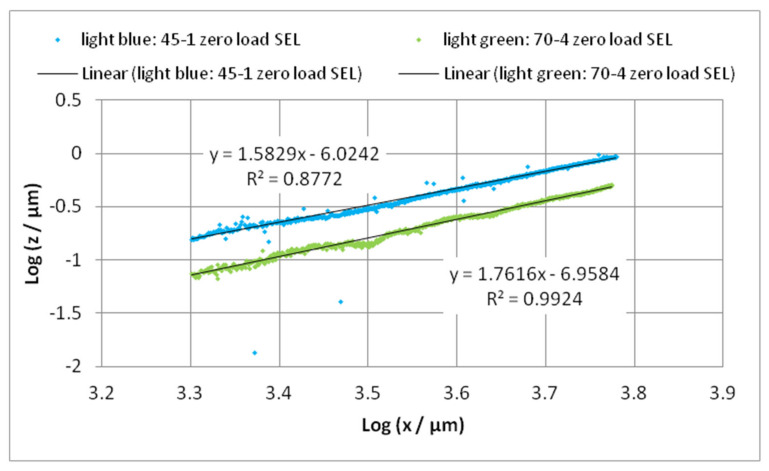
Double logarithmic representation of the zero-load bending; determination of the slope of the linear part (from 2000 µm), which is the power of the bending line.

**Table 1 sensors-22-06253-t001:** The bending *z*_w_ of cantilevers caused by their own weight calculated with Equation (4).

Nominal Thickness/µm	30	45	70	100	200
***z*_w_/µm**	−0.385	−0.171	−0.067	−0.034	−0.007

**Table 2 sensors-22-06253-t002:** Geometrical dimensions of several cantilevers ^1^.

	Thickness *h*/µm		Width *b*/µm	Length *l*/µm
Chip-No. (Old No.)	Nominal	Measured	Measured	Measured
**Chip 45-1** (6_40)	45	44.56 ± 0.4%	1040.3 ± 0.2%	6013 ± 0.2%
**Chip 45-2** (6_39)		44.27 ± 0.4%	1041.3 ± 0.4%	6028 ^2^
**Chip 45-3** (8_39)		41.00 ± 0.3%	1038.7 ± 0.4%	6028 ± 0.05%
**Chip 45-4** (10-43)		45.44 ± 0.7%	1043.8 ± 0.2%	6029 ± 0.1%
**Chip 70-1** (15-39)	70	69.21 ± 0.1%	988.0 ± 0.5%	5978 ± 0.2%
**Chip 70-2** (15-41)		69.60 ± 0.2%	989.7 ± 0.4%	6004 ± 0.1%
**Chip 70-3** (17-42)		73.55 ± 0.2%	990.7 ± 0.2%	5946 ± 0.05%
**Chip 70-4** (17-43)		73.66 ± 0.1%	992.7 ± 0.3%	5946 ± 0.05%
**Chip 100-1** (1-39)	100	95.07 ± 0.2%	944.0 ± 0.1%	5975 ± 0.02%
**Chip 100-2** (1-41)		95.02 ± 0.9%	944.7 ± 0.2%	5969 ± 0.1%
**Chip 100-3** (3-40)		94.35 ± 0.3%	945.3 ± 0.2%	5955 ± 0.2%
**Chip200-1** (19_39)	200	192.6 ^2^	732.7 ± 0.4%	5764 ± 0.1%
**Chip200-2** (21_39)		197.9 ± 0.6%	737.7 ± 0.8%	5737 ± 0.02%
**Chip200-3** (21_40)		198.0 ± 0.3%	736.0 ± 0.9%	5733 ± 0.1%

^1^ The etching process leads to slightly different dimensions when repeated. ^2^ Only the mean value is known.

**Table 3 sensors-22-06253-t003:** Young’s modulus *E* calculated from the dimensions *h*, *b*, *l* and the measured stiffness. The errors of *E* are only based on the errors of the stiffness (*E*<100> = 130 GPa is the nominal value).

Nominal Thickness *h* = 45 µm		Nominal Thickness *h* = 70 µm		Nominal Thickness *h* = 100 µm		Nominal Thickness *h* = 200 µm	
*c* N/m	*E* GPa	*c* N/m	*E* GPa	*c* N/m	*E* GPa	*c* N/m	*E* GPa
Chip 45-1		Chip 70-1		Chip 100-1		Chip200-1	
14.72 ± 0.33%	138.7 ± 0.11%	50.71 ± 1.1%	132.7 ± 1.1%	124.2 ± 0.5%	133.2 ± 0.5%	875.33 ± 1.5%	127.7 ± 1.5%
		50.79 *	133				
Chip 45-2		Chip 70-2		Chip 100-2		Chip200-2	
14.28 ± 0.4%	138.8 ± 0.4%	52.6 ± 0.1%	135.2 ± 0.1%	123.9 ± 0.3%	129.8 ± 0.3%	1002.7 ± 0.7%	130.4 ± 0.7%
14.3 *	139	52.4 *	135	123.05 *	129		
Chip 45-3		Chip 70-3		Chip 100-3		Chip200-3	
11.02 ± 0.4%	134.5 ± 0.4%	60.47 ± 0.8%	129.3 ± 0.8%	119.8 ± 1.1%	127.8 ± 1.1%	975.8 ± 1.0%	128.3 ± 1.0%
Chip 45-4		Chip 70-4					
15.16 ± 0.3%	138.4 ± 0.3%	61.42 ± 0.35%	130.7 ± 0.42%				

* Measured with instrument 2 at the PTB Braunschweig.

**Table 4 sensors-22-06253-t004:** Young’s modulus *E* calculated from the dimensions *h*, *b* and the measured stiffnesses *c* using the difference method (*E*<100> = 130 GPa is the nominal value).

Nominal Thickness *h* = 45 µm			Nominal Thickness *h* = 70 µm			Nominal Thickness *h* = 100 µm			Nominal Thickness *h* = 200 µm		
Chip	Δ*x* µm	*E* GPa	Chip	Δ*x* µm	*E* GPa	Chip	Δ*x* µm	*E* GPa	Chip	Δ*x* µm	*E* GPa
45-2	1000	134	70-1	1000	132	100-1	1000	128	200-3	1000	132
	2000	127		2000	132		2000	130		1000	128
	1000	135		1000	132		1000	131		1000	129

**Table 5 sensors-22-06253-t005:** Mean values of the zero-load bending *z*_0*E*_ resulting from the autofocus scans “cross” (Figure 6).

**Nominal Thickness**	**45** (45-1)		**70** (70-4)		**100** (100-4)		**200** (200-4)	
Offset because of “bow”/µm		0.044		0.007		0.064		0.012
Mean value *z*_0E_ (uncorr.) and *z*_0E_/µm	0.509	0.553	0.349	0.356	0.208	0.272	0.018	0.030
Standard deviation/µm	0.007		0.006		0.007		0.005	
Relative deviation/%	1.3%		1.8%		3.3%		27%	

**Table 6 sensors-22-06253-t006:** The zero-load bending of microforce standards measured by a Fizeau interferometer.

Chip No#	FC30 21-2	FC30 21-3	FC30 21-4	FC30 21-5
Nominal thickness, µm	30	30	30	30
*z*_0E_ with single-end leveling ^1^, µm	1.02 ± 0.10	1.18 ± 0.08	1.21 ± 0.10	1.18 ± 0.06
*z*_0E_ with single-end leveling ^2^, µm	1.11 ± 0.05	1.31 ± 0.06	1.28 ± 0.06	1.20 ± 0.05
*z*_0E_ with dual-end leveling, µm	0.98 ± 0.04	0.90 ± 0.05	1.03 ± 0.04	0.85 ± 0.05
*z*_0E_ with substrate leveling ^1^, µm	0.99 ± 0.05	0.91 ± 0.05	1.04 ± 0.06	0.87 ± 0.07
*z*_0E_ with substrate leveling ^2^, µm	0.97 ± 0.05	0.97 ± 0.07	1.01 ± 0.05	0.86 ± 0.07
*z*_0E_ with cross measuring, µm	0.93 ± 0.01	0.93 ± 0.01	1.00 ± 0.01	0.86 ± 0.06

^1.^ Length of ROL: 1.3 mm; ^2.^ Length of ROL: 2.5 mm.

**Table 7 sensors-22-06253-t007:** The determination of the stylus force from the stiffness and the zero-load bending, as shown in Equation (6). The error of *c* is the standard deviation, the estimation of errors of *z***_E_** and *z*_0**E**_ is described in Section 4.2.1.

Cantilever	*c*/N/m	*z*_E_/µm	*z*_0E_/µm	*z*_E_ − *z*_0E_/µm	*F*/µN
45-1	14.72	−13.436	0.938	−14.373	−211.6
	±0.05	±0.110	±0.017	±0.128	±2.6
	±0.33%			±0.89%	±1.22%
70-4	61.47	−2.931	0.502	−3.433	211.0
	±0.21	±0.014	±0.018	±0.032	±2.7
	±0.35%			±0.41%	±1.28%

**Table 8 sensors-22-06253-t008:** Determination of the stylus force by evaluating the whole profile.

Chip	*E* GPa	*b* µm	*h* µm	*I* = *bh*^3^/12 m^4^	*z*−*z_0_/x*^3^ m^−2^	3*EI* Nm^2^	*F* = *3EI*(*z*−*z_0_)/x*^3^ µN
45-1	138.7	1040	44.56	7.673 × 10^−18^	65.955	3.19271 × 10^−6^	210.6
70-4	130.7	993	73.66	3.306 × 10^−17^	16.367	1.29627 × 10^−5^	212.2

**Table 9 sensors-22-06253-t009:** Mean values of the stylus force *F* determined by the difference method from the last 1000 µm of the measured profiles using cantilevers of different thickness.

Chip		Δ*x* = 500 µm	Δ*x* = 1000 µm	Δ*x* = 2000 µm	MV
**45-1**	***F*/µN**	210.6 ± 3.4%	207.1 ± 2.6%	204.0 ± 0.9%	208.8 ± 1.2%
**70-4**	***F*/µN**	199.0 ± 6.1%	202.1 ± 3.4%	202.6 ± 3.4%	201.2 ± 1.0%

**Table 10 sensors-22-06253-t010:** The budgets of the uncertainty of *z_E_* and *z_E_*_0_ of the cantilevers 45-1 and 70-4.

		$z_{tac};z_{AF}$ µm	$\delta S_{ROL}$ µm/µm	*L* µm	$\delta z_{zero}$ µm	$\delta z_{ROL}$ µm	${\left(\frac{\Delta z}{\Delta x}\right)}_{E}$ µm/µm	*δ**x* µm	$\delta z_{lat}$ µm	$\delta z_{E};\delta z_{0E}$ µm
45-1	tactile	0.001	1.64 × 10^−7^	12075	0.0026	0.0046	−0.0092	12	−0.1102	0.1103
	zero load	0.01	6.54 × 10^−7^	8860	0.0075	0.0133	0.0004	12	0.0049	0.0173
70-4	tactile	0.001	2.13 × 10^−7^	12085	0.0032	0.0058	−0.0011	12	−0.0132	0.0144
	zero load	0.01	6.29 × 10^−7^	9050	0.0087	0.0144	0.0002	12	0.0021	0.0177

## Data Availability

No open source data exist. Data can be appropriated for demand.

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
