# Peer review of "Silicon Cantilever for Micro/Nanoforce and Stiffness Calibration"

_sensors, 2022, doi:10.3390/s22166253_

Round 1

Reviewer 1 Report

This paper presents the measurements of force-deflection characteristics using stylus.

To determine the stiffness of silicon cantilever, authors measured the tactile profile along silicon cantilever. I don't know what the merit of this method is.

Usually, non-contact method such as laser is used. It is very sensitive and simple method. The proposed method seems very time-consuming technique, too.

Also,  authors did not present the purpose of this research exactly.

In the introduction, the related researches were not sufficiently provided.

The fabrication process of Si cantilever should be presented in detail because it may affect the proposed method.

Finally, the contents is not well-organized so it is difficult to understand what authors explain.

Minor issues

1. line 133, For the measurement of force-bending-characteristics by the authors 1, ~ .  It should be changed.

2. What is the difference using instrument 1 and 2? 

3. E was used for end mark and Young's modulus. It should be modified.

Author Response

Point to point response to Review Report Form (1)

Comments and Suggestions for Authors

This paper presents the measurements of force-deflection characteristics using stylus.

To determine the stiffness of silicon cantilever, authors measured the tactile profile along silicon cantilever. I don't know what the merit of this method is.

No, the measurement of the force-deflection characteristics determines the stiffness of silicon cantilever. The measurement of the tactile profile along silicon cantilever serves the measurement of the stylus force.

Usually, non-contact method such as laser is used. It is very sensitive and simple method. The proposed method seems very time-consuming technique, too.

Just the determination of the measuring force of tactile surface profilers is the aim. Tactile profilers are widespread instruments for the measurement of surface features. The proposed method is not time-consuming. The used chip with the cantilever has to be scanned in few seconds as each other specimen for measurement of the surface profile.

Also,  authors did not present the purpose of this research exactly.

The purpose is the development of a method for measurement or quick check of stylus force of tactile surface profilers. This contains the last sentence of section 1 (Introduction) and the second last paragraph of section 5 (Conclusions).

In the introduction, the related researches were not sufficiently provided.

A lot of researches is related to cantilevers with nm- and µm-dimensions in instruments of AFM. Only a few of such papers provide informations applicable to cantilevers with µm- and mm-dimensions described in the presented paper.

The fabrication process of Si cantilever should be presented in detail because it may affect the proposed method.

The fabrication process of Si cantilevers is one of the simplest of bulk silicon microtechnique and exceeds the scale of the paper.

Finally, the contents is not well-organized so it is difficult to understand what authors explain.

This results from the fact that the main purpose of the research (measurement of the stylus force) must be described in line with a number of related problems (measurements of stiffness, of the bending profile and of zero load bending)

Minor issues

  1. line 133, For the measurement of force-bending-characteristics by the authors 1, ~ .  It should be changed.

Ok, will be canged.

  1. What is the difference using instrument 1 and 2? 

The instruments differ mainly by the stages in z-direction, see line 140.

  1. E was used for end mark and Young's modulus. It should be modified.

The Endmark E is designed by type Calibri; Young's modulus is Palatino linotype Italic.

Reviewer 2 Report

The authors propose the use of passive micromachined cantilevers as a method of force calibration.

The manuscript can be improved if the authors can clarify the following:

1. How is this calibration technique affected by changes in temperature? While electronic calibration systems that might have methods for compensation, how does one deal with changes in the operational conditions using these passive devices?

2. Can the authors comment on the repeatability using the proposed method? Approximately how many cycles of calibration can be done before wear and tear affects the mechanical properties of the cantilever?

Author Response

Point to point response to Review Report Form (2)

  1. How is this calibration technique affected by changes in temperature? While electronic calibration systems that might have methods for compensation, how does one deal with changes in the operational conditions using these passive devices?

The paper describes not the calibration of cantilevers but the method suitable for calibration. The described measurements are performed in laboratories with temperature near 20°C. The suitability of silicon cantilevers as a material measure for calibration of forces is given by the very small dependence of the Young’s modulus of silicon on the temperature:

1/E100*(dE100/dT) = -9.21138E-05 K-1           (300 K < T < 1000 K)

calculated from the temperature coefficients of the tensor of the elastic moduli of silicon [5]. This means the change of the Young’s modulus amounts 0.01% per K. In contrast to electronic systems no recalibration of a passive cantilever is necessary by use in a laboratory.

  1. Can the authors comment on the repeatability using the proposed method? Approximately how many cycles of calibration can be done before wear and tear affects the mechanical properties of the cantilever?

Effects of wear and tear of crystalline materials result from defects of the structure which can changed by temperature and mechanical stress. In the used monocrystalline electronic silicon wafers no defects (dislocations) exist and cannot created at T < 600°C. This means the silicon cantilevers show no effects of wear and tear. However if the mechanical strength is exceeded brittle fracture occurs. The bending strength of silicon cantilevers with thickness ≤ 100 µm amounts ≥ 1 GPa [5].

Reviewer 3 Report

The authors have presented a thorough work on a silicon cantilever for use in measurement of force in the micro region. The paper has also attempted to measure the Young’s modulus of silicon and the zero bending force of the cantilevers. Overall, the experiments done seem to be very detailed and multiple runs and experiments have allowed to measure the uncertainty with some level of confidence. However, there are a few major concerns:

1.       The most important aspect is that the understanding of the novelty of the paper and the methodology is a bit tough and explanation on it is vague because a lot of literature already exists about silicon cantilevers and their mathematical treatment. So, it is important to compare the work against the others. A table of comparison is highly recommended.

2.       Another important point is that the results should be compared with other modes of measurement of force that are commonly used and Young’s modulus explaining the superiority of the method and the device developed.

3.       In the methods section, if possible, all the equipment used can be listed in a short section for the benefit of the readers and for other researchers to use this work as a reference for the future.

Author Response

Point to point response to Review Report Form (3)

The authors have presented a thorough work on a silicon cantilever for use in measurement of force in the micro region. The paper has also attempted to measure the Young’s modulus of silicon and the zero bending force of the cantilevers. Overall, the experiments done seem to be very detailed and multiple runs and experiments have allowed to measure the uncertainty with some level of confidence. However, there are a few major concerns:

  1. The most important aspect is that the understanding of the novelty of the paper and the methodology is a bit tough and explanation on it is vague because a lot of literature already exists about silicon cantilevers and their mathematical treatment. So, it is important to compare the work against the others. A table of comparison is highly recommended.

The novelty is the comprehensive investigation of silicon cantilevers for use in measurement or quick check of the stylus force of tactile measuring instruments of surface topography.      A lot of literature already exists about silicon cantilevers with dimensions of nm to µm used in instruments of AFM but we found no paper dealing with cantilevers with dimensions of µm or mm used in instruments of classical metrology of surface profiles. So, a table of comparison makes no sense.

  1. Another important point is that the results should be compared with other modes of measurement of force that are commonly used and Young’s modulus explaining the superiority of the method and the device developed.

The stiffness of a cantilever can be determined by 3 methods: calculation at the base of the dimensions and a known value of Young’s modulus, extraction from measurements of intrinsic frequency and measurement of the static force- deflection-characteristic. As pointed out by Dannberg [Entwicklung eines Prüfstandes zur rückführbaren Kalibrierung von Cantilevern, Dissertation, Technischen Universität Ilmenau 2020] only the last method has a pleasable small uncertainty in the range of 1 to 4 %.

  1. In the methods section, if possible, all the equipment used can be listed in a short section for the benefit of the readers and for other researchers to use this work as a reference for the future.

We think the description of the instruments in a separate section away from the description of the measuring process is not so helpful.

Round 2

Reviewer 1 Report

I was sorry that authors did not answer the reviewer's questions properly.

The answers for the questions should be kindly provided one by one to make the reviewer understand, but I did not think that I got them sufficiently. 

It was still hard to find the purpose of this research from the revised manuscript.

Too much information (several Tables) may lead to reader's misunderstanding. 

Also, there were still many things that were not relevant to this paper, and these should be appropriately shortened and revised.

Author Response

I was sorry that authors did not answer the reviewer's questions properly.

Please, this point of review must be formulated in tangible terms.

The answers for the questions should be kindly provided one by one to make the reviewer understand, but I did not think that I got them sufficiently. 

May be, the reviewer has not received the point to point response?

It was still hard to find the purpose of this research from the revised manuscript.

The third sentence of the abstract: “The application as a material measure for the inspection of stylus forces is in the centre of investigations.”

Too much information (several Tables) may lead to reader's misunderstanding. 

An other reviewer has wished an additional table.

Also, there were still many things that were not relevant to this paper, and these should be appropriately shortened and revised.

Also this point of review must be formulated in tangible terms.

Reviewer 2 Report

No comments

Author Response

No comments - no responses are needed.

Reviewer 3 Report

Although the authors seem adamant on not putting any comparision and maybe it is okay for someone. But, do understand that it makes it so much easier to understand the novelty. Atleast you have highlighted it in the abstract so it is okay. But, do understand that now a days a lot of interdisciplinary work is being done and not everyone will be familiar with the work in the field to understand the merit of your work in comparision to others.

Author Response

No questions were formulated.

This manuscript is a resubmission of an earlier submission. The following is a list of the peer review reports and author responses from that submission.